# Vinegar-Preserved Sea Fennel: Chemistry, Color, Texture, Aroma, and Taste

**DOI:** 10.3390/foods12203812

**Published:** 2023-10-17

**Authors:** Sanja Radman, Petra Brzović, Mira Radunić, Ante Rako, Mladenka Šarolić, Tonka Ninčević Runjić, Branimir Urlić, Ivana Generalić Mekinić

**Affiliations:** 1Department of Food Technology and Biotechnology, Faculty of Chemistry and Technology, University of Split, Ruđera Boškovića 35, HR-21000 Split, Croatia; sanja.radman@ktf-split.hr (S.R.); pbrzovic@ktf-split.hr (P.B.); msarolic@ktf-split.hr (M.Š.); 2Department of Plant Science, Institute for Adriatic Crops and Karst Reclamation, Put Duilova 11, HR-21000 Split, Croatia; mira.radunic@krs.hr (M.R.); tonka.nincevic@krs.hr (T.N.R.); 3Centre of Excellence for Biodiversity and Molecular Plant Breeding, Svetošimunska Cesta 25, HR-10000 Zagreb, Croatia; 4Department of Applied Science, Institute for Adriatic Crops and Karst Reclamation, Put Duilova 11, HR-21000 Split, Croatia; ante.rako@krs.hr (A.R.); branimir.urlic@krs.hr (B.U.)

**Keywords:** *Crithmum maritimum*, apple cider vinegar, wine vinegar, alcoholic vinegar, pickles, chemical characteristics, sensory attributes, organoleptic properties

## Abstract

The aim of this study was to produce non-fermented preserved sea fennel leaves in different pickle juices prepared with apple cider vinegar, wine vinegar and alcoholic vinegar, and to compare their chemical parameters (pH, titratable acidity and salt content), organoleptic properties (color and texture parameters; volatile aromatic compound profiles) and sensory attributes. The pH of the samples ranged from 3.49 to 3.64, the lowest being in the alcoholic vinegar sample and the highest being in the wine vinegar sample, while the titratable acidity and salinity were higher in the alcoholic vinegar pickle juice than those in the other two samples. The volatile aromatic compounds of the samples were also detected. The reddish color of the wine vinegar negatively affected the sea fennel color parameters (L* and b*), and was also negatively evaluated by the panelists, while the alcoholic vinegar maximally preserved the green tones of the leaf (a*). Firmness influences the quality perceived by consumers and was therefore also tested as one of the most important parameters for evaluating the textural and mechanical properties of the different products. All sensory parameters of the sea fennel preserved in alcoholic vinegar, namely color, texture, taste, aroma and overall impression, were given the highest scores, while the sample preserved in wine vinegar received the lowest scores. The intense aroma of the wine vinegar was described as a negative characteristic (off-flavor) of the sample.

## 1. Introduction

Vegetables have high nutritional value, but are highly perishable and have a short shelf life. Therefore, their preservation has long been practiced via various methods. Pickling in pickle juice is one of the oldest and most widely used traditional methods, which ensures preservation through high acidity. The production of pickles is simple, it requires relatively simple equipment, and the final products are relatively safe and long-lasting if properly preserved.

Pickling can be performed in several ways: via anaerobic fermentation with lactic acid bacteria, via acidification with vinegar with/without the addition of salt or via a combination of these processes [1,2,3,4]. The first method is based on the activity of fermentative microorganisms that produce various metabolites that create an environment that prevents the growth and multiplication of undesirable, spoilage and pathogenic microorganisms. These metabolites lower the pH of the medium, and thus preserve the products. The second method results in non-fermented products and is based on the direct addition of acetic acid (an acidifier, usually in the form of vinegar) and other flavoring agents, followed by heat treatment (pasteurization) or cold storage in combination with the addition of preservatives [2,3,4,5]. For most products, the main preservatives used during processing are organic acids (with lactic acid and acetic acid being the most commonly used) and salt, but commercial preservatives such as sorbates, benzoates or sulfites may also be added. For non-fermented pickles, the addition of vinegar ensures a high acetic acid content, and in most cases this step is followed by pasteurization to ensure safety and prevent the product from spoiling. On the other hand, pickles produced via fermentation are not heated [1,3,4].

Salt and sugar are usually added to the pickles. Salt enhances the flavor, affects the texture and water content, and acts as a preservative by affecting water activity and reducing the solubility of oxygen [2,6]. It also helps prevent softening of tissue and assists in breaking the membranes of the raw material, allowing the diffusion/extraction of various components from the plant material into the pickle juice and vice versa [4]. For pasteurized products semi-preserved in weaker pickle juice or vinegar, salt provides some protection against microbial spoilage after the product has been opened at home [6]. Sugar, on the other hand, is added to accelerate fermentation or to make the product sweeter [1]. The addition of sugar and salt also allows the use of lower concentrations of acetic acid [3].

Sea fennel (*Crithmum maritimum* L., Apiaceae) is one of the most widespread halophyte species on the Adriatic coast. Although its leaves have been used for culinary and medicinal purposes for centuries, this valuable plant has been recently rediscovered and recognized as the star of coastal cuisine, but also as a “cash” plant for saline agriculture [7,8,9]. It stands out due to its sensory properties in terms of taste, odor and color, but also contains numerous valuable phytochemicals such as vitamin C, phenolics, unsaturated fatty acids, organic acids, minerals, etc. [10,11,12,13,14,15,16,17]. Sea fennel is used raw, cooked or preserved in various foods and dishes [8,9,18,19]. Pickled sea fennel is a product widely used in the cuisine of Mediterranean countries and usually can be found in local shops selling authentic and local traditional products. The aim of this study was to produce non-fermented pickled sea fennel using apple cider, wine and alcoholic vinegar, to evaluate the main chemical parameters of the products (pH, total acidity, salinity and volatile compounds), color and texture parameters, and to study the sensory attributes and general consumer acceptance of the products.

## 2. Materials and Methods

### 2.1. Pickling

Fresh, young and undamaged leaves of wild sea fennel collected in the area of the city of Split (Croatia) in late June 2023 were washed three times with tap water, drained and carefully dried with a paper towel. The fresh plant material was put into clean jars and immersed in pickle juice. In accordance with traditional recipes and the results of the preliminary study, three types of pickle juices were prepared using different types of vinegar: (a) apple cider vinegar (5% acetic acid; 1:5, *v*/*v*), (b) wine vinegar (6% acetic acid, 1:4, *v*/*v*) and (c) alcoholic vinegar (9% acetic acid, 1:5, *v*/*v*). All pickle juices contain salt (1%, *w*/*v*) and sucrose (granulated sugar) (1.5%, *w*/*v*). The filled and sealed jars were subjected to a pasteurization treatment (at 95 °C, for 10 min). After cooling (4 h), they were checked via lid inspection (if the center of the lid is depressed, the jar is sealed). The jars were stored at room temperature for 8 weeks before analysis.

### 2.2. Chemical Composition

Regarding the chemical composition, the pH, titratable acidity and salinity of the pickle juices were analyzed. The pH of the samples was measured using a digital pH meter (Hanna HI -2002-02 Edge, Hanna instruments Ltd., Leighton Buzzard, UK). Titratable acidity (g/100 mL) was determined using the modified AOAC method via the titration of pickle juices (10 mL) with standard sodium hydroxide (0.1 M) using phenolphthalein as the endpoint color indicator and expressed in acetic acid equivalents [20]. The salt content (g/100 mL) was determined in accordance with the modified method of Mohr by determining the chloride ion concentration by titrating the pickle juice sample (1 mL) diluted with 50 mL of distilled water with silver nitrate solution (0.1 M) [21]. All measurements were carried out in triplicate and the average values were reported.

### 2.3. Volatiles

The aromatic or volatile organic compounds (VOCs) of the pickled sea fennel were extracted via headspace–solid-phase microextraction (HS-SPME) and detected via GC-MS. HS-SPME was performed using a DVB/CAR/PDMS (divinylbenzene/carboxene/polydimethylsiloxane) SPME fiber (Agilent Technologies, Palo Alto, Santa Clara, CA, USA) conditioned in accordance with the manufacturer’s instructions before analysis. Samples (1 g) were placed in 20 mL glass vials sealed with a stainless steel lid with a polytetrafluorethylene (PTFE)/silicon septum. First, the sample was equilibrated at a temperature of 40 °C for 15 min, after which extraction was continued for 45 min. Subsequent thermal desorption lasted 6 min in the inlet set at 250 °C, from where the compounds were passed directly into the GC column. GC-MS analysis was performed using a gas chromatograph (model 8890A, Agilent Technologies) and a tandem mass spectrometer (MS) model 7000D GC/TQ. Compounds were separated on a HP -5MS UI column (30 m × 0.25 mm × 0.25 µm; Agilent Technologies). The following operating conditions were used for the gas chromatograph: a 250 °C injector temperature; a 300 °C detector temperature; a split ratio of 1:5; a column temperature program of 2 min isothermal conditions at 70 °C, followed by a temperature gradient from 70 °C to 200 °C at 3 °C/min and a further retention of 15 min. The carrier gas was helium at a flow rate of 1.0 mL/min; the ion voltage was 70 eV; the mass range was set to 30 to 300 m/z. The compounds were identified by comparing their retention indices (RI), which were based on the retention times of n-alkanes (C8-C30), with those reported in the literature (National Institute of Standards and Technology) and by comparing their mass spectra with those from the Wiley 9 (Wiley, New York, NY, USA) and NIST 17 (Gaithersburg, MD, USA) mass spectral libraries. Percent composition was calculated using the normalization method (without correction factors). Analyses were performed in duplicate and expressed as an average percentage of peak area ± standard deviation.

### 2.4. Color and Texture Parameters

The color analysis of the pickled sea fennel leaves was determined using a CIELAB color system (CR-400 Chroma Meter, Konica, Tokyo, Japan) and expressed in terms of the parameters lightness, L*; a*; b*; C and h. Standard illuminant D65 was used as a reference. The color parameter L* indicates the lightness and varies from completely opaque (0) to completely transparent (100). The value a* represents the redness and ranges from negative values for green to positive values for red. The value b* is the yellowness ranging from negative values for blue to positive values for yellow. Saturation (C) is the color intensity and hue angle (h) is the color purity, and both describe the visual impression of a color. The value C was calculated as C = [a × 2 + b × 2]^1/2^ and the hue (h) value as h = arctang b*/a*. The hue angle evaluates the different colors as follows: 0–90° red-violet, 90–180° yellow, 180–270° blue-green, and 270–360° blue [16,22]. All measurements were performed in triplicate, and the results are expressed as mean ± standard deviation.

The texture of sea fennel leaves was analyzed using a texture analyzer (TA Plus; Lloyd Instruments, Fareham, UK) equipped with a 500 N load cell and a Warner–Bratzler blade set with a rectangular slot blade. Prior to testing, the leaves were drained and brought to room temperature (25 °C). After reaching the pretest load of 0.01 N, the blade cut through the sample at a crosshead speed of 10 mm/min until the blade reached an extension of 5 mm and was then retracted to the zero position. Texture testing was carried out using Nexygen Plus 3 software. The parameters measured were hardness (N) and work of cutting (N/mm). In the software manual, hardness was defined as the maximum force during the test, while work of cutting was defined as the total work during the test. The analyses were performed in 12 replicates and the results are expressed as mean ± standard deviation.

### 2.5. Sensory Evaluation

Sensory evaluation was performed by 12 untrained panelists (aged between 28 and 48, including 6 men and 6 women) familiar with pickled sea fennel, using two methods. In the first method, samples were scored on a 5-point scale for color, texture, taste, aroma and overall impression, which includes visual, textural, taste and flavor attributes, as described by Amouroso et al. [16]. The scores were ranked as follows: 5: excellent, 4: good, 3: moderate, 2: poor, and 1: extremely poor. In addition, defects such as off-odor and mechanical damage were rated (5: none, 4: slight, 3: moderate, 2: severe, and 1: extreme). The coded samples were served at room temperature, and water was used as a palate cleanser. The arithmetic mean of the scores (*n* = 12) for each sensory attribute was used for further data processing. Although for reliable information on consumer acceptability the number of assessors should be higher, for the purpose of this study the same panelists were used for another method of sensory evaluation of the preserved sea fennel samples using a 9-point rating in a hedonic scale (the test of consumers’ preference). In this method, the scores were ordered as follows: 9: like extremely, 8: like very much, 7: like moderately, 6: like slightly, 5: neither like nor dislike, 4: dislike slightly, 3: dislike moderately, 2: dislike very much, and 1: dislike extremely.

### 2.6. Statistical Analysis

Statistical procedures were carried out in SPSS Statistics (2012; version 21.0, IBM Corp., Armonk, NY, USA). A one-way ANOVA and the general linear model (GLM) were used to investigate the significance of physicochemical, aromatic, color and texture changes in sea fennel leaves placed in three different types of pickle juice for 8 weeks of storage at room temperature. Pairwise comparisons of means were made using the least significant difference (LSD) test.

## 3. Results and Discussion

While preservation methods for various vegetables via fermentation or the direct addition of acidulant have been extensively studied, there are few studies on sea fennel products. Fermented canned sea fennel was studied by Maoloni et al. [15], while artificially acidified, pasteurized sea fennel in pickle juice was studied by Maoloni et al. [23].

Organic acids have been used for decades as additives for food preservation; acetic acid (in the form of vinegar) is commonly used in pickles at a concentration of 0.5–2% [2,5]. In this study, we used three types of commercial vinegar, apple cider vinegar, red wine vinegar and alcoholic vinegar, to prepare pickling juice solutions (at a ratio of 1:5 *v*/*v* in the case of apple cider vinegar and alcoholic vinegar and 1:4 *v*/*v* for wine vinegar), which were used to pickle sea fennel leaves.

### 3.1. Chemistry

The results for the pH, salinity and total acidity of pickle juice of sea fennel pickles after two months of storage, which were investigated in the present study, are shown in Table 1.

pH is a crucial parameter for the shelf life and safety of food products. According to Aljahani [5], the accepted standard for pickles is 4.6 or lower, which prevents the survival of most spoilage and pathogenic organisms and effectively preserves the final products. The pH of canned pickles is higher than that of fresh pickles, while the acidity and salt concentration of fresh pickles are higher. The acidity and acetic acid content of a pickled product are also important for its flavor, aroma and texture [2,24]. From the results presented in Table 1, the pH of the samples ranged from 3.49 to 3.64, being lowest in alcoholic vinegar and highest in wine vinegar. In the study by Maoloni et al. [15], the pH of the samples in most sea fennel lactic acid fermentation experiments of sea fennel shoots reached values below 3.80 after 24 h. On the other hand, Maoloni et al. [23] reported pH values of 3.85 for pickled sea fennel after 4 weeks of storage.

However, the titratable acidity of the sample preserved in alcoholic vinegar was significantly higher (2.5 g/100 mL) than that in two other samples, where it was 1.6 g/100 mL, which was expected due to the use of different acetic acid concentrations in pickle juice preparation. This was expected since the initial concentrations of acetic acid in prepared pickle juices differ (10 g/L in apple cider vinegar, 12.5 g/L in wine vinegar and 18 g/L in alcoholic vinegar). The salt content of the samples prepared with apple cider and wine vinegar was the same, while it was slightly higher (1.25 g/100 mL) when alcoholic vinegar was used.

### 3.2. Volatiles

The headspace profile of volatile organic compounds (VOCs) of the preserved sea fennel samples is presented in Table 2. The compounds were extracted via headspace–solid-phase microextraction (HS-SPME) and analyzed via gas chromatography–mass spectrometry (GC-MS).

In order to reach conclusions about the volatiles originating from the plant material and those from the vinegars, the sea fennel leaves before processing and pure vinegars were also analyzed. Briefly, 95.73% (apple cider vinegar sample) to 97.11% (alcoholic vinegar sample) of VOCs were identified. The majority of compounds originated from sea fennel (88.54% in the alcoholic vinegar sample; 90.01% in wine vinegar sample) (Figure 1) but vinegars were the source of only seven compounds detected (Table 2). 

There is a lack of information on the HS-SPME analysis of sea fennel leaves; only GC-MS analyses of EO or hydrolate are available. Terpenes dominate in sea fennel essential oils. Politeo et al. [25] reported that sabinene is the most abundant volatile compound in sea fennel leaves (51.47%) followed by limonene (36.28%) and terpinen-4-ol (5.35%). Alves-Silva et al. [26] analyzed the essential oil of the flowering aerial parts of sea fennel and the major compounds were *γ*-terpinene (33.6%), sabinene (32.02%) and thymol methyl oxide (15.7%).

The monoterpene limonene was the major compound in the HS of preserved sea fennel leaf in all of vinegars (from 43.31% in the alcoholic vinegar sample to 52.64% in the apple cider vinegar sample) followed by the monoterpene alcohol terpinen-4-ol (from 9.40% in the alcoholic vinegar sample to 12.85% in the wine vinegar sample) (Table 2). Three other terpenes, *γ*-terpinene, (*E)-β*-ocimene, and *α*-pinene, were more abundant than were other compounds. Among the compounds originating from vinegar, acetic acid was the most abundant (6.10% in the apple cider vinegar sample; 8.40% in the alcoholic vinegar sample). In addition to acetic acid, vinegar contains a variety of other secondary constituents that contribute to its flavor and that are a result of used raw material or vinegar processing steps [2]. These compounds were not detected, which is probably due to the predominance of compounds derived from the sea fennel leaves.

### 3.3. Color and Texture

Color is among the first and most important food quality parameters that attract consumers’ attention and influence their choice and opinion of the product [22]. Since the human eye can only detect minimal changes in color, the use of the CIELAB color system in the color evaluation of different samples and products is widespread, especially in industry. Therefore, the color variability of the preserved sea fennel leaves in terms of lightness (L*), greenness/redness (a*), blueness/yellowness (b*), color intensity or saturation (C) and hue angle (h) was measured and the results are shown in Table 3. The tested samples are presented in Figure 2. 

In our study, significantly higher lightness was recorded for sea fennel leaves preserved in apple cider vinegar (38.48) than for sea fennel leaves preserved in wine and alcoholic vinegar (35.30 and 36.21, respectively). Sea fennel leaves preserved in alcoholic vinegar had the highest green content (a* −1.45), and the lowest content of yellow (b* 15.28) pigment than those under the other two treatments. The color intensity (C) of sea fennel leaves preserved in alcoholic vinegar was the lowest (15.36), while the h (95.03) of those was the highest, indicating that the sea fennel leaves were generally lighter, i.e., they had a better-preserved natural green color. When treated with apple cider vinegar, there was a significant loss of green but also of yellow pigment (lowest a* and b* values). The color differences between the treatments could be closely related to the vinegar used. While the alcoholic vinegar was completely colorless, the apple cider vinegar was pale yellow and the wine vinegar was slightly reddish.

One of the key factors in the consumer acceptance and value of food, besides appearance and taste, is texture. Hardness is one of the most important parameters for evaluating the physical and mechanical properties of various foods and often influences their quality and acceptance by consumers.

It is well known that texture is determined via the moisture and fat content of the food, but also via the type and amount of structural components such as carbohydrates, proteins, hydrocolloids, etc. Thermal processes such as pasteurization, apart from their preservative effect, are often used to modify product texture, but the latter is also influenced by many other factors such as pH (with a minimum softening in the range of pH 4), salts and acids [27,28].

Therefore, when processing sea fennel leaves, special attention should be paid to the processing steps that could influence the nature of the final products, such as pasteurization (duration and temperature) and/or the composition of the soaking pickle juice (acidity and salinity). The results of our preliminary studies on preserved sea fennel were decisive for the choice of parameters used to find the optimal recipe for processing pickled sea fennel.

The textural properties of foods can be evaluated via descriptive sensory tests or instrumental analyses [28]. In this study, they were tested using a texture analyzer, and the results obtained, such as hardness and work of cutting of the product, are shown in Table 2. The work of cutting, described as the total work achieved during the test, was significantly (*p* < 0.05) higher for the sea fennel leaves preserved in wine vinegar than it was for the other two samples. The hardness of the pickled sea fennel leaves was not affected by the type of pickle juice.

Texture is one of the most important parameters for pickled sea fennel, because leaves that are too soft, tender and tough are undesirable. The fennel samples pickled in alcoholic vinegar had an average hardness and were characterized by a lower work of cutting, which was also positively evaluated by the panelists in the sensory evaluation, while the hardness of the samples pickled in wine vinegar was considered a negative attribute of the product. According to Brenes et al. [27], fruit softening is induced via the addition of acid, the effect being stronger with lactic acid than with acetic acid. In their study, the effect of acid on fruit hardness was greater at lower pHs and consequently at higher acid concentrations. According to the LSD test, the lowest values for work of cutting were found in the alcoholic vinegar samples, which had, significantly (*p* < 0.01), the highest titratable acidity. On the other hand, MacFeeters et al. [29] reported that salt softens low-acidity vegetables such as cucumbers. The LSD test also showed that the alcoholic vinegar sample had significantly (*p* < 0.01) the highest salt content in the pickle juice, indicating that the salt was probably extracted from the plant material, which could have an effect on softening and reducing the chopping values.

### 3.4. Sensory Evaluation

Descriptive sensory analysis was used to describe the sensory attributes of preserved sea fennel with three different vinegars (Figure 3 and Table 4).

In this test, panelists rated the color, texture, taste, aroma and overall impression of the samples using a five-point scale and also noted any noticeable defects (off-odors and mechanical damage). As can be seen from the results, the sea fennel preserved with alcoholic vinegar received the highest values for all tested attributes. The highest value was given for taste (4.25 ± 0.25), color, texture and overall impressions, which were evaluated with an average score of 4.17, while the aroma of the product was evaluated to have a slightly lower score (3.83 ± 0.52). Sea fennel preserved with apple cider vinegar received high scores for color and texture, while the score for the aroma of the sample was significantly lower (3.25 ± 0.87). Few panelists described this sample as too acidic. Sea fennel preserved in wine vinegar, on the other hand, had the lowest values for all parameters (all below 4.00). The highest was for texture (3.92 ± 0.29), and the lowest was for the taste (3.00 ± 0.50) of the sample. The greatest differences between the samples were found for color, alcoholic (4.17) > apple (4.08) > wine vinegar (3.58), which was probably due to the coloration of the pickle juice samples. The color of the wine vinegar affected the color of the fennel leaves, so the split parts of the leaves and shoots were colored red, which was rated negatively by the panelists. The intense aroma of the wine vinegar was also described as a negative characteristic (off-flavor) of the sea fennel samples.

This study also included a test of consumer preference using the hedonic scale rating (Figure 4). The results show that only the sea fennel preserved in alcoholic vinegar received the highest rating (extremely like), while the sample in red wine vinegar was the only one with a low rating (dislike very much). In addition, the sample in alcoholic vinegar was evaluated mostly via high grades on a hedonic scale (like extremely, like very much and like moderately). These results are in accordance with the descriptive sensory tests of the preserved sea fennel samples.

## 4. Conclusions

Sea fennel is one of the most widespread halophytes on the Adriatic coast, with a valuable nutritional composition and a number of health-promoting ingredients. Its leaves, harvested in the summer months, have been used for centuries in the culinary tradition and in the food industry due to their special sensory properties (taste, smell and color) and are used in various foods raw, cooked or preserved. However, in the Croatian culinary tradition they are mainly prepared as fresh-packed pickles and used in salads. This study investigated the chemical, color, texture and organoleptic properties of sea fennel preserved with different vinegars, as well as overall consumer acceptance. The results showed that alcoholic vinegar received the highest score in the sensory evaluation, while the sample preserved with wine vinegar, which is mainly used in the traditional preservation of sea fennel in the Croatian coastal region, received the lowest scores and the negative perception of consumers, mainly due to the strong aromatic vinegar notes and negative attributes for color and firmness.

## Figures and Tables

**Figure 1 foods-12-03812-f001:**
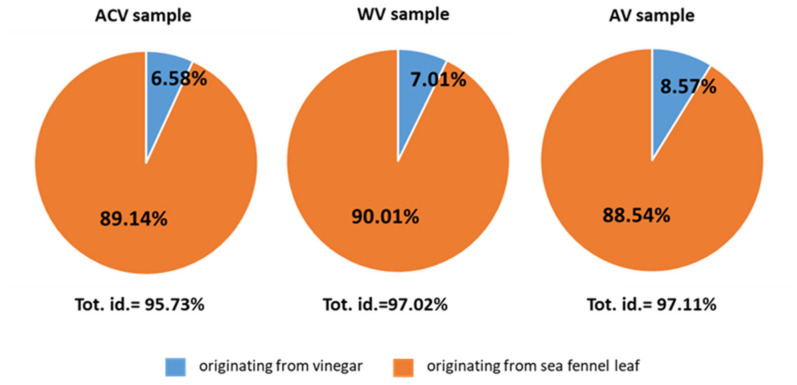
Distribution of origin of compounds in preserved sea fennel samples. ACV—apple cider vinegar; WV—wine vinegar; AV—alcoholic vinegar; tot. id.—total identified.

**Figure 2 foods-12-03812-f002:**
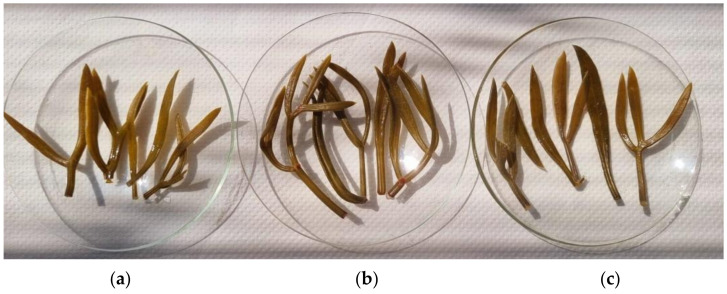
Samples of pickled sea fennel leaves in (**a**) apple cider vinegar, (**b**) wine vinegar and (**c**) alcoholic vinegar.

**Figure 3 foods-12-03812-f003:**
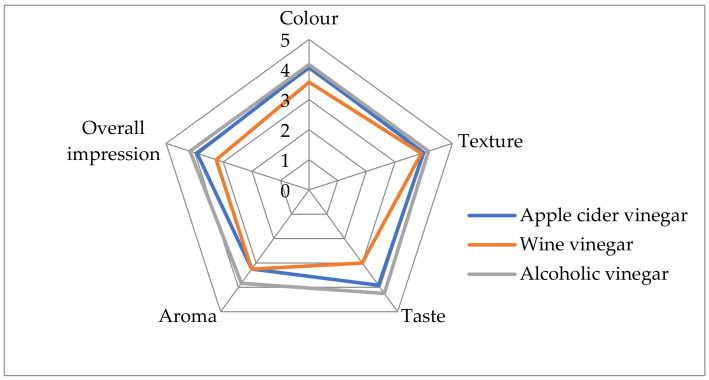
Sensory profiles of sea fennel preserved in pickle juices prepared using different vinegars (5: excellent, 4: good, 3: moderate, 2: poor, and 1: extremely poor).

**Figure 4 foods-12-03812-f004:**
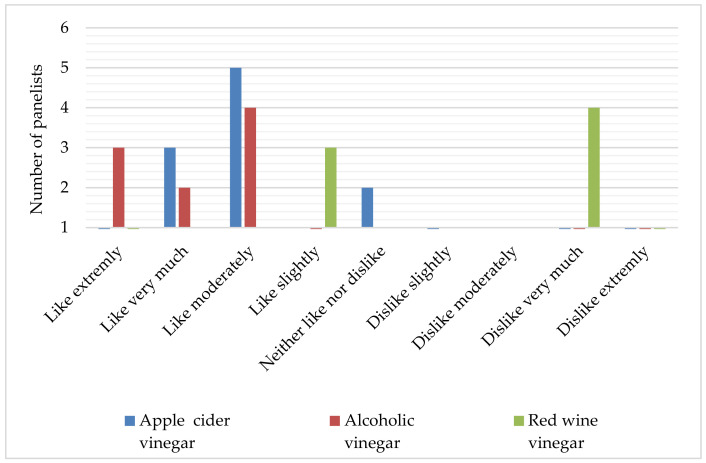
Hedonic scale rating of sea fennel preserved in pickle juices prepared using different vinegars.

**Table 1 foods-12-03812-t001:** pH values, salt content and total acidity of the sea fennel samples preserved via the addition of three types of vinegars.

	Apple Cider Vinegar (1:5, *v*/*v*)	Wine Vinegar (1:4, *v*/*v*)	Alcoholic Vinegar (1:5, *v*/*v*)	*p*-Value
pH value	3.55 ± 0.05	3.64 ± 0.09	3.49 ± 0.06	0.09
Titratable acidity (g/100 mL)	1.62 ^a^ ± 0.01	1.65 ^a^ ± 0.01	2.48 ^b^ ± 0.01	<0.01
Salt (g/100 mL)	1.16 ^a^ ± 0.02	1.16 ^a^ ± 0.02	1.25 ^b^ ± 0.01	<0.01

Mean values within a row with different letters in superscript differ significantly (*p* < 0.05).

**Table 2 foods-12-03812-t002:** The volatile organic compounds (VOCs, %) of preserved sea fennel leaf in apple cider, wine and alcoholic vinegar.

No.	RT (min)	Compound	RI	RI Library	Apple Cider Vinegar (1:5, *v*/*v*)	Wine Vinegar (1:4, *v*/*v*)	Alcoholic Vinegar (1:5, *v*/*v*)	*p*-Value
1.	1.84	Acetic acid *	<800	600	6.10 ^a^ ± 0.30	6.41 ^a^ ± 0.08	8.40 ^b^ ± 0.54	<0.01
2.	4.99	α-Pinene	947	939	4.69 ± 0.26	5.17 ± 0.34	5.04 ± 0.35	0.36
3.	5.34	Camphene	962	961	0.06 ± 0.03	0.07 ± 0.03	0.10 ± 0.05	0.46
4.	5.41	(*E*)-Hept-2-enal	965	956	0.07 ± 0.02	0.04 ± 0.01	-	0.21
5.	5.56	Benzaldehyde *	971	961	-	-	0.02 ± 0.00	-
6.	6.01	Sabinene	987	978	0.20 ^a^ ± 0.00	0.14 ^b^ ± 0.01	0.10 ^c^ ± 0.01	<0.01
7.	6.19	6-Methylhept-5-en-2-one	992	987	0.01 ± 0.00	-	-	-
8.	6.31	β-Myrcene	996	994	1.16 ^a^ ± 0.09	1.69 ^b^ ± 0.10	2.06 ^c^ ± 0.13	<0.01
9.	6.47	(*E,Z*)-Hepta-2,4-dienal *	1002	999	0.01 ± 0.00	-	-	-
10.	6.64	Octanal	1008	1005	0.09 ± 0.02	0.05 ± 0.02	0.04 ± 0.02	0.10
11.	6.74	α-Phellanderene	1012	1105	0.40 ^a^ ± 0.05	0.38 ^a^ ± 0.01	0.48 ^b^ ± 0.02	<0.05
12.	7.10	α-Terpinene	1023	1020	0.17 ^a^ ± 0.00	0.43 ^b^ ± 0.03	0.90 ^c^ ± 0.06	<0.01
13.	7.60	Limonene	1037	1035	52.64 ^a^ ± 2.21	43.75 ^b^ ± 2.22	43.31 ^b^ ± 2.58	<0.01
14.	7.76	(*E*)-β-Ocimene	1045	1040	5.99 ^a^ ± 0.04	8.30 ^b^ ± 0.41	8.50 ^b^ ± 0.42	<0.01
15.	8.05	(*Z*)-β-Ocimene	1056	1051	0.59 ^a^ ± 0.04	0.94 ^b^ ± 0.01	1.23 ^c^ ± 0.01	<0.01
16.	8.45	γ-Terpinene	1066	1064	6.26 ^a^ ± 0.08	8.04 ^b^ ± 0.953	7.81 ^b^ ± 0.52	<0.01
17.	9.44	α-Terpinolene	1097	1089	0.28 ^a^ ± 0.00	0.44 ^b^ ± 0.02	0.86 ^c^ ± 0.05	<0.01
18.	9.83	Linalool	1103	1104	0.02 ± 0.00	0.02 ± 0.00	0.01 ± 0.00	0.71
19.	10.00	(3*E*)-6-Methylhepta-3,5-dien-2-one *	1108	1107	0.08 ^a^ ± 0.00	0.06 ^b^ ± 0.00	0.07 ^b^ ± 0.00	<0.01
20.	10.28	*p*-Mentha-1,3,8-triene	1117	1119	-	-	0.04 ± 0.01	-
21.	10.31	2-Phenylethanol *	1117	1117	0.05 ^a^ ± 0.03	0.32 ^b^ ± 0.01	0.07 ^a^ ± 0.02	<0.01
22.	10.60	(3*E*,5*E*)-2,6-dimethylocta-1,3,5,7-tetraene	1126	1134	0.29 ^a^ ± 0.04	0.21 ^b^ ± 0.00	0.17 ^b^ ± 0.00	<0.02
23.	10.92	(*Z*)-Alloocimene	1136	1130	1.22 ^a^ ± 0.08	2.04 ^b^ ± 0.14	2.23 ^b^ ± 0.15	<0.03
24.	11.52	β-Terpineol	1151	1159	0.09 ^a^ ± 0.00	0.12 ^b^ ± 0.01	0.11 ^c^ ± 0.00	<0.04
25.	11.94	4-prop-1-en-2-ylcyclohexene *	1161	1161	0.33 ± 0.01	0.16 ± 0.00	-	-
26.	12.08	(*Z*)-Non-2-enal	1161	1162	0.04 ± 0.01	0.02 ± 0.00	0.02 ± 0.00	0.23
27.	12.36	*p*-Mentha-1,5-dien-8-ol	1171	1171	0.02 ± 0.01	0.02 ± 0.00	0.02 ± 0.00	1.00
28.	12.88	Terpinen-4-ol	1183	1178	10.03 ^a^ ± 0.32	12.85 ^b^ ± 0.95	9.40 ^a^ ± 0.64	<0.01
29.	13.34	α-Terpineol	1193	1190	1.18 ^a^ ± 0.02	1.68 ^b^ ± 0.07	1.37 ^c^ ± 0.06	<0.02
30.	14.47	(*E*)-Carveol	1222	1222	0.36 ^a^ ± 0.03	0.21 ^b^ ± 0.01	0.16 ^b^ ± 0.00	<0.03
31.	14.94	(*Z*)-Carveol	1234	1232	0.09 ^a^ ± 0.02	0.06 ^ab^ ± 0.01	0.03 ^b^ ± 0.01	<0.05
32.	15.13	Thymol methyl ether	1239	1237	0.01 ± 0.00	0.01 ± 0.00	0.01 ± 0.00	0.42
33.	15.49	Carvone	1248	1246	1.16 ^a^ ± 0.12	0.58 ^b^ ± 0.04	0.60 ^b^ ± 0.04	<0.01
34.	15.84	α-Ionene	1256	1255	0.02 ^a^ ± 0.00	0.05 ^b^ ± 0.00	0.02 ^a^ ± 0.00	<0.05
35.	16.03	2-Phenylethyl acetate *	1261	1260	0.02 ^a^ ± 0.00	0.06 ^b^ ± 0.01	0.02 ^a^ ± 0.00	<0.01
36.	16.21	(*E*)-Dec-2-enal	1265	1260	0.07 ^a^ ± 0.01	0.04 ^b^ ± 0.00	0.03 ^c^ ± 0.00	<0.01
37.	17.54	*o*-Thymol	1295	1294	0.06 ^a^ ± 0.00	0.05 ^a^ ± 0.00	0.03 ^b^ ± 0.00	<0.01
38.	22.70	β-Caryophyllene	1421	1428	0.13 ^a^ ± 0.00	0.35 ^b^ ± 0.02	0.35 ^b^ ± 0.02	<0.01
39.	23.29	γ-Elemene	1436	1433	0.10 ^a^ ± 0.04	0.15 ^a^ ± 0.05	0.44 ^b^ ± 0.05	<0.01
40.	23.38	α-Bergamotene	1438	1436	0.14 ± 0.03	0.17 ± 0.03	0.20 ± 0.04	0.30
41.	23.48	Aromadendrene	1441	1442	0.04 ± 0.01	0.07 ± 0.01	0.06 ± 0.01	0.50
42.	25.27	α-Curcumene	1484	1485	0.18 ± 0.02	0.19 ± 0.00	0.18 ± 0.00	0.58
43.	25.38	(*E*)-β-Ionone	1487	1485	0.02 ± 0.00	0.02 ± 0.00	0.02 ± 0.00	0.87
44.	25.76	α-Zingiberene	1496	1496	0.17 ^a^ ± 0.00	0.38 ^b^ ± 0.01	0.79 ^c^ ± 0.03	<0.01
45.	26.30	β-Bisabolene	1510	1509	0.26 ^a^ ± 0.02	0.32 ^b^ ± 0.00	0.44 ^c^ ± 0.01	<0.01
46.	26.88	β-Sesquiphellanderene	1526	1526	0.41 ^a^ ± 0.03	0.49 ^a^ ± 0.03	0.70 ^b^ ± 0.04	<0.01
47.	27.52	Selina-3,7(11)-dien	1543	1543	0.02 ^a^ ± 0.00	0.03 ^b^ ± 0.00	0.09 ^c^ ± 0.00	<0.01
48.	28.11	Germacrene B	1558	1558	0.04 ^a^ ± 0.00	0.07 ^b^ ± 0.00	0.22 ^c^ ± 0.02	<0.01
49.	29.14	Caryophyllene oxide	1584	1583	0.15 ^a^ ± 0.01	0.15 ^a^ ± 0.01	0.11 ^b^ ± 0.00	<0.01
50.	29.44	α-Guaiol	1591	1595	0.21 ± 0.02	0.23 ± 0.00	0.20 ± 0.01	0.14

* originated from vinegar; RI—retention index. Results are expressed as mean ± standard deviation (SD); the average of two results was used as the third repetition in the ANOVA. Mean values within a row with different letters in superscript differ significantly (*p* < 0.05).

**Table 3 foods-12-03812-t003:** Color and texture parameters of the sea fennel persevered via the addition of apple cider, wine and alcoholic vinegar.

	Apple Cider Vinegar (1:5, *v*/*v*)	Wine Vinegar (1:4, *v*/*v*)	Alcoholic Vinegar (1:5, *v*/*v*)	*p*-Value
Color parameters				
L*	38.48 ^a^ ± 0.33	35.30 ^b^ ± 0.32	36.21 ^b^ ± 0.33	<0.01
a*	−0.27 ^a^ ± 0.01	−0.84 ^ab^ ± 0.01	−1.45 ^b^ ± 0.02	<0.05
b*	20.62 ^a^ ± 0.60	23.03 ^b^ ± 0.59	15.28 ^c^ ± 0.61	<0.01
C	20.64 ^a^ ± 0.58	23.05 ^b^ ± 0.57	15.36 ^c^ ± 0.56	<0.01
h	90.88 ^a^ ± 0.43	91.22 ^a^ ± 0.43	95.03 ^b^ ± 0.42	<0.01
Texture parameters				
Load at Maximum Load (N)	8.22 ± 1.79	10.47 ± 0.45	8.56 ± 1.58	0.21
Work of shear (N/mm)	14.19 ^a^ ± 3.52	19.62 ^b^ ± 1.72	13.91 ^a^ ± 1.05	<0.05

Results are expressed as mean ± standard deviation; mean values within a row with different letters in superscript differ significantly (*p* < 0.05).

**Table 4 foods-12-03812-t004:** Sensory parameters of the sea fennel preserved via the addition of apple cider, wine and alcoholic vinegar.

	Apple Cider Vinegar (1:5, *v*/*v*)	Wine Vinegar (1:4, *v*/*v*)	Alcoholic Vinegar (1:5, *v*/*v*)	*p*-Value
Color	4.08 ± 0.38	3.58 ± 0.29	4.17 ± 0.52	0.25
Texture	4.00 ± 0.66	3.92 ± 0.29	4.17 ± 0.28	0.81
Taste	3.92 ^a^ ± 0.80	3.00 ^b^ ± 0.50	4.25 ^a^ ± 0.25	<0.05
Aroma	3.25 ± 0.87	3.25 ± 0.25	3.83 ± 0.52	0.44
Overall impression	3.92 ± 0.63	3.25 ± 0.50	4.17 ± 0.38	0.16

Results are expressed as mean ± standard deviation; mean values within a row with different letters in superscript differ significantly (*p* < 0.05).

## Data Availability

The data presented in this study are available on request from the corresponding author.

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
