# Peer review of "Vinegar-Preserved Sea Fennel: Chemistry, Color, Texture, Aroma, and Taste"

_foods, 2023, doi:10.3390/foods12203812_

Round 1

Reviewer 1 Report

The manuscript describes different organoleptic parameters in sea fennel preserved using three types of vinegar.

The presentation is clear and the experimental design is correct. However, there are some aspects that could be modified to improve the quality of the work:

-        Is there any commercial or traditional sea fennel preserved in vinegar commonly accessible? I think it is important to consider this, and specially regarding to the untrained panel. I mean, it is not the same if the panellists are familiarized or not with a product.

-        Check the word panellist instead of “panelist”

-        Line 345 check alcoholic>apple>alcoholic one should be “wine” I guess

-        The format of table 2 is not correct, the logo of the journal is on top and the position should be landscape. I am not able to review this table because I just see partially the results.

-        It is not clear how you distinguish which compounds are from vinegar or sea fennel. Is by the literature? Previous work? Did you analysed the vinegars previously?

Author Response

Reviewer #1

The manuscript describes different organoleptic parameters in sea fennel preserved using three types of vinegar.

The presentation is clear and the experimental design is correct.

Thank you for the positive remark.

However, there are some aspects that could be modified to improve the quality of the work:

-        Is there any commercial or traditional sea fennel preserved in vinegar commonly accessible? I think it is important to consider this, and specially regarding to the untrained panel. I mean, it is not the same if the panellists are familiarized or not with a product.

Pickled sea fennel is a local product that is widely used in the traditional cuisine of Dalmatia, but also in other Mediterranean countries (e.g. Italy, Greece, etc.). In Croatia, pickled sea fennel can be found in local shops selling authentic and local traditional products. Pickled sea fennel is usually produced by family businesses according to their own recipes, while there are no SMEs or companies producing it on a larger scale (such as the Rinci company in Italy). This has been added in the paper. Furthermore, all panellists involved in the study were familiar with the product.

-  Check the word panellist instead of “panelist”

It has been changed.

-  Line 345 check alcoholic>apple>alcoholic one should be “wine” I guess

It has been changed.

-  The format of table 2 is not correct, the logo of the journal is on top and the position should be landscape. I am not able to review this table because I just see partially the results.

It has been changed.

-   It is not clear how you distinguish which compounds are from vinegar or sea fennel. Is by the literature? Previous work? Did you analysed the vinegars previously?

Yes, vinegars and sea fennel leaves were analysed separately, but these data were not presented. We have included this information in the manuscript to make it clear.

Reviewer 2 Report

The publication titled: " Vinegar preserved sea fennel: Chemistry, color, texture, aroma and taste" is well described in terms of carrying out the methodology.

My question is:

Is preserved sea fennel a locally popular product? Is it known worldwide or only locally?

Author Response

The publication titled: " Vinegar preserved sea fennel: Chemistry, color, texture, aroma and taste" is well described in terms of carrying out the methodology.

My question is: Is preserved sea fennel a locally popular product? Is it known worldwide or only locally?

We would like to thank the reviewer for the positive comment. Pickled sea fennel is a local product that is widely used in the traditional cuisine of Dalmatia, but also in other Mediterranean countries (e.g. Italy, Greece, etc.). In Croatia, pickled sea fennel can be found in local shops selling authentic and local traditional products. Pickled sea fennel is usually produced by family businesses according to their own recipes, while there are no SMEs or companies producing it on a larger scale (such as the Rinci company in Italy, which is our project partner).

Reviewer 3 Report

The paper investigates am imnteresting use of a perennial wild plant. However, some corrections are needed:

- the term "brine" is only used for salt-water combinations, the vinegar-containing preservation uses "pickle juice". Please, correct.

- Why did the Authors use different acetic acid concentrations is their preserves? This makes comparison very subjective.

-Also, why did not the Authors use white wine vinegar instead of the red wine vinegar?

-L142: reference needed

Generally, 3 replicates are needed for one-way ANOVA tests. Please, justify the use of this test with only 2 replicates (GC-MS).

- What was the number of replicates in texture analysis?

Author Response

Reviewer #3

The paper investigates am interesting use of a perennial wild plant. However, some corrections are needed:

- the term "brine" is only used for salt-water combinations, the vinegar-containing preservation uses "pickle juice". Please, correct.

It has been corrected.

- Why did the Authors use different acetic acid concentrations is their preserves? This makes comparison very subjective.

We thank the reviewer for this comment. Yes, we have used vinegars with different contents of acetic acid and in different proportions, as we have mentioned in the main text in a few places. The applied concentrations in the use of apple juice vinegar and wine vinegar were mostly used in traditional recipes for pickled sea fennel, while the applied concentration in the case of alcoholic vinegar was the result of our preliminary research, as this type of vinegar is not usually used in traditional preparation.

-Also, why did not the Authors use white wine vinegar instead of the red wine vinegar?

In this study we were following the common methods of the sea fennel preservation in pickle juices used in Dalmatian traditional cuisine where mostly (homemade) red wine vinegar is used. Also, white wine vinegar is hard to find in the Croatian market, but results of this study affected plans for our new study where distilled wine vinegar will be used for preservation.

  • L142: reference needed

It has been added.

- Generally, 3 replicates are needed for one-way ANOVA tests. Please, justify the use of this test with only 2 replicates (GC-MS).

Yes. In case of GC-MS results only two replicates were performed and the third value was calculated as a mean value. This is now stated below the table. All other assays/methods were performed with min 3 replicates.

- What was the number of replicates in texture analysis?

In texture analysis we had 12 replicates (by selecting younger (6) and older (6) leaves in order to make the results more reliable). This is now stated in the paper.